# Packaging Evaluation Approach to Improve Cosmetic Product Safety

**Benedetta Briasco [1], Priscilla Capra [1], Arianna Cecilia Cozzi [1], Barbara Mannucci [2] and Paola Perugini [1,3,\***

[1] Department of Drug Sciences, University of Pavia, Via Taramelli 12, 27100 Pavia, Italy; benedetta.briasco@gmail.com (B.B.); priscilla.capra@unipv.it (P.C.); arianna.cecilia01@gmail.com (A.C.C.)
[2] C.G.S (Centro Grandi Strumenti), University of Pavia, Via Taramelli 12, 27100 Pavia, Italy; barbara.mannucci@unipv.it
[3] Etichub s.r.l, Academic Spin-Off, University of Pavia, Via Taramelli 12, 27100 Pavia, Italy
**\*** Correspondence: paola.perugini@unipv.it; Tel.: +39-3-8298-7174

**Abstract:** In the Regulation 1223/2009, evaluation of packaging has become mandatory to assure cosmetic product safety. In fact, the safety assessment of a cosmetic product can be successfully carried out only if the hazard deriving from the use of the designed packaging for the specific product is correctly evaluated. Despite the law requirement, there is too little information about the chemical-physical characteristics of finished packaging and the possible interactions between formulation and packaging; furthermore, different from food packaging, the cosmetic packaging is not regulated and, to date, appropriate guidelines are still missing. The aim of this work was to propose a practical approach to investigate commercial polymeric containers used in cosmetic field, especially through mechanical properties' evaluation, from a safety point of view. First of all, it is essential to obtain complete information about raw materials. Subsequently, using an appropriate full factorial experimental design, it is possible to investigate the variables, like polymeric density, treatment, or type of formulation involved in changes to packaging properties or in formulation-packaging interaction. The variation of these properties can greatly affect cosmetic safety. In particular, mechanical properties can be used as an indicator of pack performances and safety. As an example, containers made of two types of polyethylene with different density, low-density polyethylene (LDPE) and high-density polyethylene (HDPE), are investigated. Regarding the substances potentially extractable from the packaging, in this work the headspace solid-phase microextraction method (HSSPME) was used because this technique was reported in the literature as suitable to detect extractables from the polymeric material here employed.

**Keywords:** safety evaluation; polyethylene; packaging; mechanical properties

---

## 1. Introduction

Packaging can be defined as an economical means of providing presentation, protection, identification, information, containment, convenience, and compliance for a product during storage, carriage, and appearance until the product is consumed. Packaging must provide protection against climatic conditions and biological, physical, and chemical hazards and must be economical. The package must ensure adequate stability of the product throughout the shelf life [1].

In recent decades, the interest of research and industry towards plastic packaging, both environmentally friendly and safe for the consumer, has exponentially grown. In the cosmetic and pharmaceutical packaging field, one of the most used plastic materials is polyethylene (PE), a thermoplastic resin obtained by polymerization of ethylene.

As a numerical example, the worldwide production capacity of PE is estimated to be 79,106 metric tons per year. Of this amount about 21,106 tons are low-density PE (LDPE), 22,106 tons linear LDPE (LLDPE), and the remaining 36,106 tons is high-density PE (HDPE).

All types of polyethylene are semi-crystalline polymers. Their densities and melting temperatures decrease with the increase of ramification. Many hundreds of grades of PE, differing in their properties, are actually available [2].

PE possesses good chemical stability [3–5]. The mechanical properties are dependent on the molecular weight and on the degree of chain branching. With increasing density, the barrier properties increase as well as the stiffness, hardness, and strength, as a result of the higher crystallinity. At the same time, there is a decrease in the impact resistance, toughness, resistance to stress cracking, cold resistance, and transparency [2].

Furthermore, polyethylene can be produced from renewable resources and it is readily recyclable if it has not been coated with other materials [6].

Blown containers from LDPE are used as packaging in the pharmaceutical and cosmetic industries as well as for food, toys, and cleaning agents. The most important application area of HDPE is the production of containers and injection-molded articles [2].

Despite the excellent characteristics of this polymer as packaging material, both plastic and its additives used in the production process can migrate from the packaging to the content over time as a result of an increase in temperature, mechanical stress, or aging. Like in the food field, the presence of plastic components or additives in cosmetics, if not properly controlled, can affect the organoleptic properties of the product, or its safety, if the levels exceed the legislated or toxicological values [7].

Furthermore, in contrast to glass or metal packaging materials, polymeric packaging are permeable at different degrees to small molecules like gas, water vapor, and to other low-molecular weight compounds like aromas, flavors, and additives present in the formulation; this is an important point, as contamination from external environment could cause reactions within the contained product (oxidation of lipids, degradation of actives, etc.) or the absorption of ambient vapor or liquid could cause an increase of polymer plasticization, resulting in a decrease in mechanical properties [8].

In particular, PE it is able to retain large amounts of nonpolar compounds, such as most of the volatile molecules, because of its polyolefin nature: this phenomena, known as aroma scalping, causes a loss of aroma content and or/aroma imbalance. On the other hand, other plastic materials (e.g., ethylene-vinyl alcohol copolymer, EVOH) are medium to poor water barrier plastics and their hydrophilic nature promotes the sorption of large amounts of water, which results in plasticization of the polymers and the subsequent loss of mechanical and barrier properties [9].

Evidence in literature show that changes in mechanical behavior causes changes on the barrier properties [10]. These kind of modifications in packaging can greatly affect the safety of consumers. In fact, it is well known that some substances can migrate from packaging to the formulation, but it is not well disseminated; yet, the knowledge about the influence of packaging mechanical changes on product safety would be improved. For example, the presence of microcracks can modify oxygen permeability and thus lead to a degradation of substances in the formulation, like preservative, reducing their activity.

For this reason, in the development of a cosmetic product safety assessment, besides the packaging raw materials information issue, other aspects related to packaging functionality should be evaluated, like possible interactions between material and product in relation to primary packaging [11].

In fact, packaging made from the same starting polymeric material but with different additives or produced by different manufacturing processes, although apparently similar, can interfere differently with the content, causing unwanted reactions on the consumer [12]. Recently, a new preservative ingredient was placed on the market to be used as an additive in the preparation of "active" packaging composed of glass beads in which silver ions are dispersed. This material received a positive opinion from the Scientific Committee on the Consumer Safety (SCCS) [13]. It is clear that any change, also

mechanical, of this kind of packaging, will affect in a decisive way the release of the preservative in a cosmetic product and consequently influence the safety of the finished product.

Compatibility tests should be performed on the product, once transferred to the final container. The container-content relationship should be explored for all the packaging materials, as the final quality of the goods is always the result of a delicate balance between these two components.

Despite the importance of these aspects, there is too little information about the possible chemical-physical interactions between formulation and packaging, because, differing from food packaging, the cosmetic one is not regulated and, to date, appropriate guidelines are still missing. However, with Regulation 1223/2009 coming into full entry force, among the voices of the Cosmetic Product Safety Report of the Product Information File (PIF), a section pointing out "Impurities, traces, information about the packaging material" has become obligatory.

This work aims to propose a protocol to characterize final packaging for underlining possible critical issues in order to assure a completely safe product to consumers.

In particular, next to analysis of the extractables, of which a lot of methods and protocols are present in literature [14–16], this work focuses on the mechanical analysis step since, as said before, changes in mechanical properties could provoke alterations of packaging performance, like barrier properties, with a consequent risk for the product's integrity.

As an example of application, a study conducted on two types of polyethylene with different densities is reported.

A simple experimental design, in order to minimize the number of trials, was employed [17,18]. Polyethylene containers were filled with standard formulations and submitted to different degradation tests (photostability test and accelerated stability test) to mimic stress conditions that products can meet during their shelf life, according to European guidelines for stability tests on cosmetic products.

Standard monophasic formulations (pH 2 and pH 10) were used, in order to carry out the test in extremes conditions.

After this treatment, the samples were analyzed by tensile test, to verify possible changes of mechanical properties. "Bone-shape" specimens, obtained from empty and filled bottles [19], were analyzed with a tensile machine until their break, obtaining stress-strain curve. The comparison between treated and untreated materials permitted the underlining of any mechanical change.

Afterwards, an extraction method was used in order to detect all the potentially extractable substances.

## 2. Materials

Packaging materials, the object of this study, were commercial containers of 250 mL capability: HDPE bottles and LDPE tubes obtained from different suppliers. The thicknesses of containers are around 500 μm and 1 mm for LDPE and HDPE, respectively.

The filling solutions were set up with the following substances: potassium chloride, 37% hydrochloric acid, borax, and potassium hydroxide drops, all provided by CARLO ERBA reagents (Cornaredo, MI, Italy).

## 3. Experimental

The proposed approach foresees different steps.

### 3.1. Provision of Data

The first step is the collection of all data regarding the considered packaging.

Companies operating in the cosmetic industry provide information about packaging for the CPSR (Cosmetic Product Safety Report), for example, the food grade certificate and test reports according to the Regulation (EC) No. 1935/2004 on Food Contact Materials [20]; the declaration/certificate of compliance according to Annex IV of Regulation (EU) No. 10/2011 (plastic materials and articles) [21]; the composition, with the specification/technical data for each raw material, based on knowledge of the process for manufacturing the raw material (origin of substance, production process, synthesis

route, extraction process, solvent used, etc.) and with a physical and chemical analysis of possible impurities in raw materials and, if necessary, in the final product (e.g., nitrosamines); and the SVHC (substances of very high concern) declaration/certificate and test report to comply with REACH regulations (packaging being considered an article under REACH).

The comparison with the requirements of food packaging could be useful because the food grade of packaging is mentioned in several EU cosmetic guidelines; there are migration tests and limits and a positive list of allowed monomers and additives. However, some substances are not included in the Union list, but they may be present in the plastic layers of plastic materials or articles, like non-intentionally added substances and additives for polymerization; furthermore, in food packaging, different from cosmetic field, colorants are not of concern and there are some substances that are allowed in Food, but regulated in EU Cosmetic Regulation (e.g., hydroquinone, phenoxyethanol, etc.).

### 3.2. Experimental Design

In order to maximize the information while reducing the number of analyses, an appropriate experiment design (screening design) has to be developed.

In this study, a simple full factorial design was chosen to investigate the effect of three experimental factors on two response parameters. The results of mechanical tests, such as the variation of stress and the percentage of elongation at break point of containers, compared to non-treated empty ones, were chosen as response parameters. In fact, we have already demonstrated that these parameters can be good indicators of any change occurring in the mechanical behavior of polymeric materials [22]. The three factors of interest were varied on two levels according to the experimental plan showed in the Table 1. The density of polyethylene (low or high density), the pH of contained solutions (2 and 10), and the kind of treatment (accelerated aging and solar simulated irradiation) were chosen as factors, to determine the influence of these parameters on mechanical properties of polyethylene used as packaging material in the pharmaceutical and cosmetic field.

**Table 1.** Investigated experimental factors and levels experimental design.

| Experimental Factors | Level | |
|---|---|---|
| | **−1** | **1** |
| Density of polyethylene | LDPE | HDPE |
| Buffer pH | 10 | 2 |
| Treatment | 30 days climatic chamber | 24 h solar box |

The order of the experiments was randomized to avoid any bias. Statistical calculations were carried out using the software StatGraphics (Statpoint Technologies, Warrenton, VA, USA).

### 3.3. Degradation Testing Procedures

The HDPE and LDPE containers (object of this work) were numbered, weighed, and washed according to a standard washing procedure [19]. Afterwards, 10 bottles for each polymer filled with standard solutions were used for each degradation test:

- Photostability test by simulating UV-visible ray irradiation using SUNTEST XLS +II (Atlas®, URAI, Assago, MI, Italy) for 24 h;
- Accelerated stability test by incubation in climatic room (ClimaCell 111 MMM) at 40 °C with 75% Relative Humidity (R.H.) for 30 days.

SUNTEST instrument was set up in according to standard European procedures, with the following parameters:

- Time: 4 h corresponding to 192 h solar light;

- Irradiation control: 300–800 nm;
- Irradiation (W/m$^2$): 750;
- Room temperature: 35 °C;
- Black standard temperature (BST): 45 °C.

Photostability test was performed in according to Colipa guidelines about cosmetic products [23]. At least three specimens were obtained from each bottle to carry out mechanical and morphological analyses in triplicate.

### 3.4. Mechanical Test

The investigation of the mechanical properties of the bottles was performed using a tensile machine, AGS 500ND (Shimadzu Corporation, Kyoto, Japan) equipped with a 500 N load cell; the test was performed using a strain rate, specific for each material, evaluated by preliminary trials:

- LDPE: 5 mm/min
- HDPE: 10 mm/min

Five "bone-shape" specimens were obtained from each container; the feature of the specimens followed the principles of the European Standard EN ISO 527 [24], suitably modified for bottle containers [19]. Briefly, an optimized dog bone shape obtained by punchcutting was used. This design was developed in order to obtain a localized stress region 3 mm width and thick. Wall thickness distributions for each sample were measured at 3 different points using a digital microscope Duratool model BW1008-500x (Farnell element14 Trade Counter, Leeds, UK). The section of each sample was calculated from thickness and width using a suitable software program (micromeasure vers. 1.2).

Samples were kept under constant temperature (23 °C) and humidity (52% R.H.) for a week until tension tests started and during the entire test time.

This procedure permitted the obtainment of a stress versus strain curve. From each set of results, it was possible to estimate the tendency of materials to oppose to deformation, and to evaluate the curve profile in elasticity regime, the elongation percentage in elasticity regime, and the absolute elongation elasticity.

A critical analysis and comparison of parameters derived from diagrams allowed a qualitative but also a quantitative assessment of any significant change that occurred in the packaging due to interactions between the material they are made of and the conditions or substances with which they are in contact.

### 3.5. Extractables' Analysis

The next step aims to obtain and interpret data from a controlled extraction's study starting from the several methods proposed in the literature.

In this work the headspace solid-phase microextraction (HSSPME, fiber: PDMS 100 micron, Supelco, Sigma-Aldrich, Gallarate, MI, Italy) was the extraction method considered for obtaining information about extractable substances from packaging.

Briefly, 500 mg of polymer was put into a vial and the HSSPME conditions used were the following: fiber: PDMS 100 micron (Supelco); adsorption temperature: 90 °C; extraction time: 60 min; desorption temperature: 250 °C; desorption time: 4 min, 30 s.

After extraction, for the identification of compounds a gas chromatography-mass spectrometry (GC-MS, Termo Scientific Trace DSQ II, Fisher Scientific Italia, Rodano, MI, Italy) was used. The GC conditions were the following: column: Restek Rtx-5MS, 30 m × 0.25 mm ID × 0.25 μm; gradient: 60 °C for 4.5 min, 20 °C/min until 280 °C, 280 °C for 5 min; injector: PTV 250 °C, split time 4.5 min, split flux 10 mL/min; gas: He, constant flux 1 mL/min; transfer line: 270 °C.

The MS conditions were: source: 250 °C; ionizing mode: EI 70 eV; scansion mode: full Scan; scansion range: 50–650 amu; scansion rate: 870 amu/s.

After analyses, a search in the spectra library, using databases like NIST/EPA/NIH Mass Spectral Library (Wiley Registry of Mass Spectral Data 8th Edition) with Search Program (MSSP) (Data Version: NIST 2008, *Software* Version 2.0) was performed in order to identify all substances recovered in the sample.

## 4. Results and Discussion

The safety assessment of a cosmetic product can be successfully carried out only if the safety assessor can obtain all information concerning the product, including the specific area of application (face, mucosa, periocular area, etc.), the people for whom the product is intended (baby, elderly people, etc.), and the conditions of use, but it is extremely important also to evaluate the hazard deriving from the use of the designed packaging.

Furthermore, commercial packaging is varies widely and it is very difficult to have complete information about it. For this reason, it is very important to define a general protocol that every manufacturer can apply, modifying it in a suitable way to its own formulation-packaging system for the development of an "in house" stability test.

Following the protocol developed in this study, it is possible to evaluate both the behavior of container itself and the possible interactions between content and container in order to ensure the quality of product and the safety for consumers.

This study case, in particular, focuses on the evaluation of one of the most used plastic packaging materials, polyethylene, to understand which are the most influential factors that could cause variations in their properties, as a starting point to extend the knowledge in this field.

After finding all the information about these packaging materials, the second step aims to evaluate the mechanical properties, designed as behavior to tensile testing, of final containers. In particular, adapted "bone-shape" specimens [19] were obtained from LDPE and HDPE bottles and then analyzed with a tensile machine.

Here parameters obtained from the tensile test are shown and discussed in order to make a comparison between the different materials.

During the tensile test, the specimen presents five basic stages, resulting in the five areas of a typical stress-strain curve:

- Elastic behavior: this corresponds to the first phase of material deformation; deformations that occur during this phase are reversible, so if at this stage the applied stress is stopped there are no residual deformations of the specimen, which restores its initial length. In this phase the elongation is directly proportional to the load (in the stress-strain diagram it is represented by a straight portion);
- Continuing the tensile test, it adopts a more linear behavior; this step is called the yield point and it corresponds to a fall of the strength of the material due to the formation of "microcracks" within the material. The yield corresponds to the initial part of the plastic behavior;
- Plastic behavior: in this phase there are both elastic (reversible) and plastic (permanent) deformation; this means that if resetting the load during this phase, there will be residual deformations associated with the contribution of plastic deformation, for which the specimen will have a greater length than at the start of the test;
- During the test, there is a localized deformation of the specimen, for which a small part of the specimen quickly decreases the area of its cross-section; this is called *necking phase* and it characterizes the descending part of the stress-strain curve;
- After necking there is the specimen *break*, which occurs in correspondence with the so-called breaking load, which corresponds to the maximum stress that the specimen can withstand;
- The reported graphs in Figures 1 and 2 show, as an example, a different mechanical behavior depending on the considered material, according to the UNI EN ISO 527 [24].

As it can be seen, the mechanical behavior of these two polymers is greatly different, in terms of elongation percentage and stress (MPa); so it is not numerically possible to compare one material with the other. For this reason, every change in mechanical properties has been evaluated, comparing each material untreated with itself after treatment.

Furthermore, it is important to underline that the approach described in this work can be successfully employed to evaluate modification of packaging during aging or during contact with the packed formulation in order to define the shelf life of the product or any interactions between formulation and the packaging.

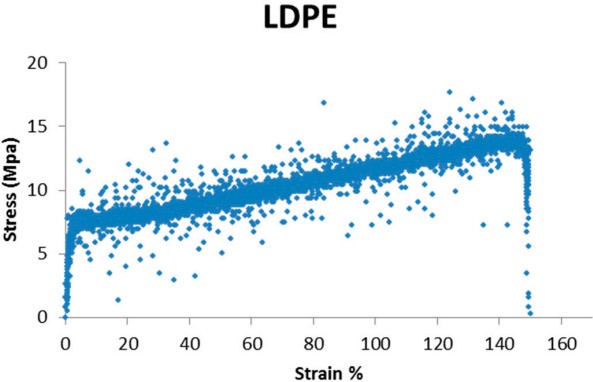

**Figure 1.** Mechanical behavior of low-density polyethylene (LDPE).

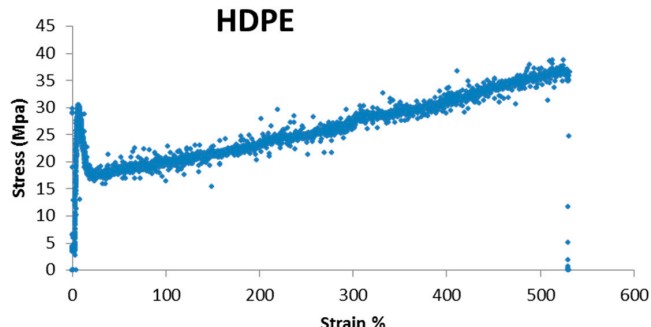

**Figure 2.** Mechanical behavior of high-density polyethylene (HDPE).

For this purpose, mechanical properties of empty and filled bottles, before and after stress testing procedures, were investigated. The stress-strain curve profile is useful to compare specimens subjected to environmental and chemical stress.

HDPE presents major strength, maybe due its linear structure, that makes the polymer more resistant, while LDPE presents a greater ability to stretch, with a lower stress value.

Results of tensile tests for different materials are reported in Tables 2 and 3, in terms of tensile stress and strain at break.

**Table 2.** Results obtained by mechanical analyses for low-density polyethylene (LDPE) containers.

| LDPE | Tensile Stress at Break ($\sigma B$) (MPa) * | Tensile Strain at Break ($\varepsilon t B$) (%) * | Δ Tensile Stress at Break (%) * | Δ Tensile Strain at Break (%) * |
|---|---|---|---|---|
| Empty | 21.30 | 150.97 | - | - |
| pH 2 sun 24 h | 17.44 | 122.62 | −18.09 | −18.78 |
| pH 2 chamber 30 days | 22.45 | 189.07 | 5.41 | 25.24 |
| pH 10 sun 24 h | 18.42 | 148.95 | −13.52 | −1.34 |
| pH 10 chamber 30 days | 21.80 | 189.20 | 2.35 | 25.33 |

* S.D. $\leq$ 10%.

**Table 3.** Results obtained by mechanical analyses for HDPE containers.

| HDPE | Tensile Stress at Break (σB) (MPa) * | Tensile Strain at Break (εtB) (%) * | Δ Tensile Stress at Break (%) * | Δ Tensile Strain at Break (%) * |
|---|---|---|---|---|
| Empty | 29.65 | 391.70 | - | - |
| pH 2 sun 24 h | 26.47 | 399.06 | −10.74 | 1.88 |
| pH 2 chamber 30 days | 24.88 | 331.21 | −16.10 | −15.44 |
| pH 10 sun 24 h | 25.31 | 325.23 | −14.65 | −16.97 |
| pH 10 chamber 30 days | 23.62 | 289.35 | −20.33 | −26.13 |

\* S.D. ≤ 10%.

Observing the values, it can be said that for LDPE there is a general reduction of the yield stress at break point. The major reduction is observable for samples treated with irradiation, regardless of the type of solution contained. So, the light has the bigger influence on material changes; this influence is exacerbated by extreme pH.

Also, regarding HDPE, we can observe that there are some changes in stress and elongation at break. The bigger variation can be observed for the samples treated in climatic chamber. It can be underlined that the container filled with the pH 10 solution has undergone the bigger changes.

Results are very interesting and they agree with literature data. In fact, it is well known that PE polymers are quite stable to degradation depending of their molecular weight, but it is also known that UV irradiation and thermal exposure can increase surface hydrophilicity of these polymer [25].

Furthermore, in all final PE packaging available in the market, antioxidants and stabilizers, in smaller or bigger amount, are present. The presence of these substances products containing PE become susceptible to degradation and subsequent oxo-biodegradation. They cause initiation and propagation of free radical chain reactions taking place in the presence of atmospheric oxygen, which leads a polymer to gradually reduce its molecular weight [26,27]. These processes cause a change in the hydrophilicity of a polymer surface, that can be more susceptible to extreme pH.

Here the Pareto Charts and the Factor Means Plots of statistical analysis of the mechanical test's results, obtained by the simple screening experimental design described above, are reported in Figures 3 and 4.

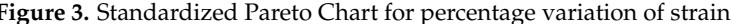

**Figure 3.** Standardized Pareto Chart for percentage variation of strain.

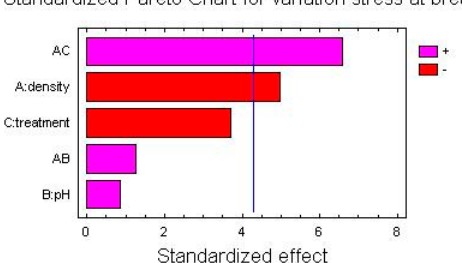

**Figure 4.** Standardized Pareto Chart for percentage variation of stress at break.

As it is possible to see from the graphs, the only factor that has a significant influence on the mechanical variations after treatment is the density of polyethylene, both regarding the variation of percentage elongation and the variation of the stress at break point.

For both for the variation of percentage elongation and for the variation of the stress at break point, the interactions between two factors—the density of the polymer and the kind of treatment (UV-vis irradiation and climatic chamber)—are significantly influential.

The main effect represents the average result of varying one factor at a time from low to high and keeping the other one constant. The interaction term shows changes in the response when both factors are varied concurrently, as this is possible to observe in the Figures 5 and 6 below reported.

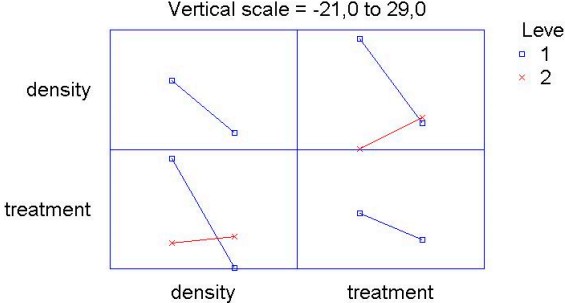

**Figure 5.** Factors Means Plot for percentage variation of strain.

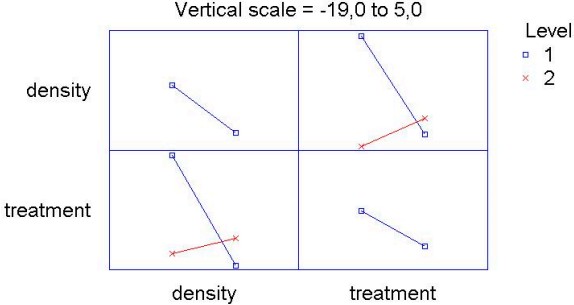

**Figure 6.** Factors Means Plot for variation of stress at break.

The considered extraction method was headspace solid-phase microextraction (HSSPME). After extraction, for the identification of compounds a gas chromatography–mass spectrometry was used. Figure 7 shows an example of the chromatogram obtained by GC/MS.

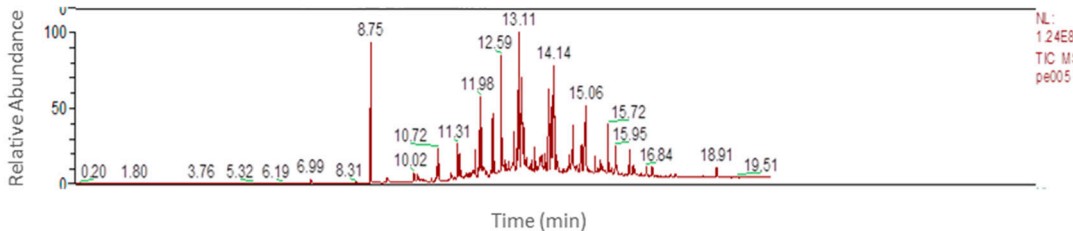

**Figure 7.** Chromatograms obtained with headspace solid-phase microextraction (HSSPME) on untreated LDPE.

The deconvolution of the chromatographic peaks leads to the identification of more than 100 substances. Many of these substances are linked to the bleeding of stationary phase of the

chromatographic column to the SPME fiber coating, and to characteristic analytes also present in blanks used as references. By eliminating the interfering peaks, a list of compounds that can be identified as extractable that were released from the analyzed polymer can be obtained. In this way it is possible to split the substances into several categories, as reported in the Table 4. From the analysis of chromatographic profiles of extraction process and of relative percentage of the different substances present in the packaging it is possible:

- To choose the better packaging for the specified cosmetic product;
- To define which substance has to be quantitatively evaluated in the final cosmetic product as leachable after stability and interaction studies.

**Table 4.** Categories of extractable type released from PE polymer.

| Extractable Type | Example |
| --- | --- |
| Initial ingredients | Antioxidants (e.g., Terbutylphenol, Irganox), additives (phtalaths), amides (exadecanammide) |
| Impurities related to processing | Oligomers, residual solvents, esters (miristyl miristate), siloxane |
| Degradation products of the polymer | Fragments of saturated and unsaturated hydrocarbons, ketones, acids |

During compatibility testing it is also possible to detect products adsorbed by the formulation contained in the packaging material.

The data show that the sample obtained from head space microextraction (HSSPME) is representative, and it also identifies numerous nonpolar organic compounds, even the most significant polar substances.

## 5. Conclusions

This work aims to provide necessary tools and a practical approach to evaluate commercial polymeric containers used in cosmetic packaging in order to assure the safety of the finished product.

In fact, it is well known that packaging can greatly affect the safety of the product by both losing its barrier property and containing substances potentially harmful for the consumer, especially for products for children or containing sunscreens.

Despite the importance of this aspect, there is too little information about the possible chemical-physical modifications of the packaging itself during aging or about the interactions between formulation and packaging.

The correct approach involves the provision of all possible information about the packaging material from suppliers' data sheet and from literature; then, an appropriate design of experiment has to be successfully used in order to obtain relevant indications minimizing the number of trials that must be carried out in order to perform an effective safety evaluation of the finished packaging used and of the interaction between each couple packaging-formulation.

Actually, the main problem is related to the actual composition of the packaging at the end of the production process. For this reason, it is essential to collect information about the container and not only the polymer raw materials used in the packaging production.

In this work the results of mechanical tests are chosen as predictive system's parameters, but this kind of approach can be used also for describing other system's parameters, for example the viscosity or other characteristics of the contained product.

After mechanical analysis, it is important to perform also an extractables' analysis; in this case the used technique was the headspace solid microextraction (HSSPME), since, compared to other techniques used in preliminary studies, this one allows the definition of almost the total extraction profile of the analyzed material.

The reported study case regards two types of polyethylene containers with different densities, HDPE and LDPE; the commercial containers made of these materials were treated in extreme conditions of pH and accelerated aging, in order to evaluate which factors have the most influence on the mechanical properties of these materials.

This work has shown that the most influential factor is the density of polyethylene, but also that the interaction between the kind of polyethylene and the kind of treatment has significant influence on the mechanical answer of the material in comparison with the same untreated material.

So, these polymers cannot be considered as completely inert and stable. Some particular conditions (for example heat, UV radiation, and humidity) may alter the chemical, physical and mechanical properties of these polymeric materials.

In conclusion, it would be very important to apply this kind of experimental approach in the development phase of a new cosmetic product before its introduction into the market.

**Author Contributions:** Paola Perugini and Priscilla Capra designed the study. Benedetta Briasco and Arianna Cecilia Cozzi performed the mechanical experiments, Barbara Mannucci and Benedetta Briasco performed the extractable experiments. Paola Perugini, Priscilla Capra and Benedetta Briasco analyzed the results and wrote the manuscript.

**Conflicts of Interest:** The authors declare no conflict of interest.

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
