# Peer review of "Packaging Evaluation Approach to Improve Cosmetic Product Safety"

_cosmetics, doi:10.3390/cosmetics3030032_

Reviewer 1 Report

Regarding the Materials and Methods section:

2.2 The table number is missing in line 150 of page 4.

2.3 How many containers were selected for assessment?

2.4 How many replicates did you do in the degradation assay?

Results section:

The legend of table 3 needs to be corrected.

The results obtained for LDPE for the Δ tensile strain at break should be explained (table 2). What is the significance of the increase in this parameter after treatment in the chamber at both pH 2 and pH 10?

More detail regarding the results extractable's analysis should be given, for instance, examples of the chromatograms obtained with each analytical technique.

Author Response

 I modified the paper in according to your observations. I added also more details on the results and a chromatogram for analytical technique used in this work.

Reviewer 2 Report

I find the topic of the manuscript to be relevant and commend the authors for taking this subject on.  However, I do not feel the manuscript accomplishes its objective, which is to bring good science and sound decision to packaging evaluations performed in the cosmetic industry. 

 Firstly, the title in in conflict with the body of the manuscript.  Thus while the title stresses product safety, the bulk of the Discussion addresses physical properties that affect performance, not safety.  Moreover, the portion of the manuscript that considers chemical assessment addresses one aspect of the assessment, which is generating the extract.  A complete evaluation of packaging safety would also need to consider how the extract is to be tested and how the resulting test data is to be interpreted.

 I found the Introduction to be much more lengthy then was necessary to introduce the issue being addressed.  It contained many facts, such as the production capacity of PE, that were not relevant to the manuscript's major focus. Additionally, the Introduction contained study-related details that are best placed elsewhere in the manuscript.

 An important outcome of the manuscript is the designation of a standard extraction matrix.  When choosing from among several candidate methods, the authors use "number of compounds extracted" as their criterion. This not not seem to be an appropriate or rigorous criterion and thus I am not sure the authors have chosen the proper method.

 Lastly, I do not find the method's descriptions to be precise enough.  While the authors refer to extraction methods in general terms such as Soxhlet and IPA/W, they do not specify any of the operational details associated with the specific methods they used.

Specific comments are as follows:

 Line 55 contains the statement that "PE is FDA compliant".  As the FDA does not qualify materials, this statement is not accurate.

Line 56 suggests that just because a material is implanted into the human body it must be safe.  I note that safety as a device does not imply safety in other applications.

As I noted previously, Section 2.6 should describe the extraction methods in greater detail.  In order for the reader to evaluate the authors' choice of standard extraction process, they must know how the authors' performed their extractions.  For example, the description of the IPA/water extraction, which I assume was a sealed vessel extraction, should include the extraction time and temperature, the amount of material extracted and the volume of the extracting solution.  Additionally, the proportion of IPA to water must be specified.  A similar level of detail should be used for the other extraction methods. 

Also in Section 2.6, I note that the authors used GC as their analytical method. While GC is an appropriate method, another method (LC), is also typically employed in extractables assessment, as many extractables cannot be detected by GC.  The authors' decision of what technique to use may be based on an incomplete data set.  

I do not understand the term "eliminating the interfering peaks" in line 294.  What are these interfering peaks and where are they coming from?

in line 296, the authors list Irganox as an additive but it is more properly termed an antioxidant.

In Table 4, I do not understand the classes of compounds the authors' use to define the columns in the Table.   

Author Response

Following your suggestions we've  modified the title, we've revised the english language.

Furthermore we've added more details in experimental methods, especially for the extraction method.

We've explained better the connections between the mechanical analyses and the safety of final product.

Finally we have modified the Table 4 and we have introduced the Figure 7 in order to be more clear.

Reviewer 3 Report

Line 56: Need citation

Lines 82–85: It is incomplete sentence. I cannot get idea from this sentence. “Different packaging…”

Line 112: This experiment was designed to test with two simulants (pH 2 and pH 10). But in my opinion, most of cosmetic is about pH 7–10. The most influent factor from cosmetic to polymer properties could be a lipid/oil/wax from emulsifier or perfume. Which is able to penetrate and alter the polymer matrix by acting as a plasticizer.

I think, to make this research more realistic and applicable to industries, the study with fatty simulant need to be investigated since acidic simulant is not a good cosmetic representative.

Line 159: About material: This research didn’t mention about the study of material for their packaging/container. There is no any data show about the contained additive in each bottle type. Moreover, one condition for stability test is under sun light. I think, the way to say that the most influent factor to the interaction between cosmetic and packaging is a polymer type (i.e., density)
(as line 328 in conclusion). They need to show that both LDPE & HDPE sample has everything similar…

For example:

(1) Adding UV stabilizers or not;

(2) The thickness of bottle’s wall is same or not because it is influence to the penetration of sun light to product (may be light acts as accelerator for product-packaging interaction) as well as the penetration of product to polymer matrix.

Author Response

In genaral we have revised the english language and we have explained methods with more details.

In particular:

Line 56: we have eliminated the FDA note.

Lines 82-85: we have modified the sentence

Line 112: about the choice of pH formulation, we have chosen these pH that exist in cosmetic formulation, for example peeling solution or hair products because we thought that pH with heat or UV irradiation could strongly interact with material changing the hydrophilicity of a polymer surface, that can be more susceptible to  extreme pH . Obviously it is important to analyse the effect of lipids or others materials too.

Line 159: we have added some details but you are right and we have added also a sentence  about the problem related to the actual composition of the packaging at the end of the production process. For this reason it is essential to collect information about the container and not only the polymer  raw materials used in the packaging production. 

Round  2

Reviewer 2 Report

I thank the authors for effectively adressing my previous comments.  I am fine with the manuscript except the last sentence that has been added to the abstract, which specifically is "'HSSPME was used because this technique was reported in the literature as suitable to detect almost all extractables from the polymeric material here employed".  This statement is inaccurate as it well known in the E&L community that mutiple orthogonal analytical techniques are required to "detect amost all extractables".  Thus a statement that implies that one technique can accomplish this objective is overly optimistic.  I would ask that the authors change this statement to read "'HSSPME was used because this technique was reported in the literature as suitable to detect extractables from the polymeric material here employed".     

Author Response

Dear Reviewer,

I've change the sentence in the abstract as you required.

Reviewer 3 Report

For stability & mechanical test, the result would be completely perfect and useful, if the authors include the blank sample (empty bottle) into the matrix for all condition of storage test. So, you can fully interpret whether the content has influence to the mechanical properties or only the irradiation and heat that impact to the bottle's properties.

Author Response

Dear reviewer,

thank you for your comment. In this work, the results of the experimental design need to be related to a single and univocal blank in order to be comparable. For this reason, wanting to apply this experimental design, it wasn’t possible to insert also empty treated containers in the results.

Furthermore, as already mentioned regarding the extractables’ paragraph, this work wanted to propose a protocol usable by companies, in order to establish their own in-house compatibility studies between formulation and chosen packaging. For this reason we propose a simple experimental design to limit the number of trials to perform. 

However, this research is going on by my research group; in particular different containers made of PE and other polymers are been studying and  the data related to further tests (included tests on the empty and treated packaging) will be inserted in another paper.